# Molecular Cloning and Analysis of an Acetyl-CoA C-acetyltransferase Gene (*EkAACT*) from *Euphorbia kansui* Liou

**DOI:** 10.3390/plants11121539

**Published:** 2022-06-09

**Authors:** Meng Wang, Zhe Zheng, Zheni Tian, Hao Zhang, Chenyu Zhu, Xiangyu Yao, Yixin Yang, Xia Cai

**Affiliations:** 1Key Laboratory of Resource Biology and Biotechnology in Western China, Ministry of Education, Northwest University, Xi’an 710069, China; wangmengss501@163.com (M.W.); z13653402708@163.com (Z.Z.); tzn2734914487@163.com (Z.T.); 18737310533@163.com (H.Z.); b18292802915@163.com (C.Z.); yxy940385143_nwu@163.com (X.Y.); yangyx_0122@163.com (Y.Y.); 2Medical Experiment Center, Shaanxi University of Chinese Medicine, Xianyang 712046, China

**Keywords:** *Euphorbia kansui* Liou, acetyl-CoA acetyltransferase gene (*AACT*), terpenoids, abiotic tolerance

## Abstract

Terpenoids are the largest class of natural products and are essential for cell functions in plants and their interactions with the environment. Acetyl-CoA acetyltransferase (AACT, EC2.3.1.9) can catalyze a key initiation step of the mevalonate pathway (MVA) for terpenoid biosynthesis and is modulated by many endogenous and external stimuli. Here, the function and expression regulation activities of AACT in *Euphorbia kansui* Liou (EkAACT) were reported. Compared with wild-type Arabidopsis, the root length, whole seedling fresh weight and growth morphology of *EkAACT*-overexpressing plants were slightly improved. The transcription levels of *AtAACT*, *AtMDC*, *AtMK*, *AtHMGR*, and *AtHMGS* in the MVA pathway and total triterpenoid accumulation increased significantly in transgenic Arabidopsis. Under NaCl and PEG treatment, *EkAACT*-overexpressing Arabidopsis showed a higher accumulation of total triterpenoids, higher enzyme activity of peroxidase (POD) and superoxide dismutase (SOD), increased root length and whole seedling fresh weight, and a decrease in the proline content, which indicated that plant tolerance to abiotic stress was enhanced. Thus, AACT, as the first crucial enzyme, plays a major role in the overall regulation of the MVA pathway.

## 1. Introduction

Terpenoids are the largest class of natural secondary metabolites and contain more than 70,000 different molecules [1]. These molecules are synthesized in mitochondria, plastids, nonorganellar cytoplasm, endoplasmic reticulum (ER) and the specialized vacuoles by the mevalonic acid (MVA) and methylerythritol-4-phosphate (MEP) pathways [2]. As vital secondary metabolites, terpenoids play important roles in many aspects of plant life, including growth, development, metabolism, and defense against biotic and abiotic stresses [1,3,4,5,6,7]. In addition, terpenoids have important economic value and are widely used in rubber production as well as the preparation of drugs, nutraceuticals, flavors, fragrances, pigments, agrochemicals and disinfectants [8]. In higher plants, all types of isoprenoids are synthesized using a five-carbon (C5) skeleton as a common precursor; the precursor is often isopentenyl diphosphate (IPP) and its double-bond isomer dimethylallyl diphosphate (DMAPP) [9]. According to the number of five-carbon units, terpenoids are divided into polyterpenoids, sesquiterpenoids, triterpenoids and sterols, which are synthesized in the cytosolic MVA pathway, and diterpenes, tetraterpenes, and monoterpenes, which are derived via the plastid MEP pathway [10]. The MVA pathway, which can produce antioxidant molecules, was proven to be crucial for maintaining the redox state and increasing fitness under abiotic stress.

In plants, the MVA biosynthesis pathway contains a series of enzymes that can catalyze acetyl-coenzyme A (COA) to form corresponding terpenoids [11,12]. The cytosolic MVA pathway starts with acetyl-CoA and progresses through the catalysis of a series of enzymes, including AACT, 3-hydroxy-3-methylglutaryl-CoA synthase (HMGS), 3-hydroxy-3-methylglutaryl-CoA reductase (HMGR), MVA kinase (MK), phospho-MVA kinase (PMK) and MVA diphosphate decarboxylase (MDC) [10]. Acetyl-CoA forms isopentenyl diphosphate (IPP), which can be converted into dimethylallyl diphosphate (DMAPP), which is a common precursor for the synthesis of terpenes (sterols, sesquiterpenes, ubiquinones, dolichols) [10,13]. Acetyl-CoA acetyltransferase (acetoacetyl-CoA thiolase, AACT), which is also called synthetic thiolase and belongs to thiolase II, catalyses the condensation of two acetyl-CoA to form acetoacetyl-CoA; this step is the first enzymatic step in the MVA biosynthesis pathway [14,15]. As a vital starting enzyme, AACT has been cloned from plants including *Arabidopsis thaliana* [16,17], *Helianthus annuus* [14,18], *Hevea brasiliensis* [19], *Bacopa monnieri* [20], *Houttuynia cordata* [21], *I**sodon rubescens* [22] and *E**uphorbia helioscopia* [23]. In addition, it was reported that in *Ganoderma lucidum*, an overexpression of the *GlAACT* gene can improve the transcription of related genes in the MVA pathway and the accumulation of triterpenes [24]. Furthermore, in *Medicago sativa*, the salinity tolerance was better in *MsAACT*-overexpressing transgenic plants than in empty vector transformed plants, which showed that AACT, as a vital regulatory enzyme in isoprenoid biosynthesis, is involved in the abiotic stress adaptation [15]. Nevertheless, further studies are needed to explore the vital regulatory function of AACT in the MVA biosynthesis pathway and adaptation of plants to abiotic stress.

*Euphorbia kansui* Liou, a Chinese medicinal material in *Euphorbia* of Euphorbiaceae, has important economic value [25,26]. It has been used as an herbal remedy for ascites and cancer and has a wide spectrum of antiviral activities. Terpenoids are considered as the major active metabolites in *E. kansui*, including diterpenes and triterpenes [27]. In the present study, to further explore the role of *EkAACT* in terpenoid biosynthesis and plant tolerance to abiotic stresses, an *AACT* gene from *E. kansui* was cloned and then overexpressed in wild-type (WT) Arabidopsis. By analyzing WT and transgenic Arabidopsis under salt and drought stress, it was found that *EkAACT* overexpression increased triterpene accumulation and the transcription level of triterpene biosynthesis genes in Arabidopsis, as well as salt and drought-stress resistance. These studies provide guidance for further analyses of the regulation of terpenoid biosynthesis and provide green methods for improving the production of bioactive compounds.

## 2. Results

### 2.1. EkAACT Cloning and Characterization

To study the function of *E. kansui* thiolase II, the full-length coding sequence was obtained using sequence-specific primers. The full-length CDS of the *EkAACT* gene is 1098 bp long and encodes a 365 amino acid polypeptide. The protein molecular weight is 38.28 kDa, and the isoelectric point (pI) is 6.74.

An NCBI-BLASTP search indicated that the amino acid sequence of the EkAACT protein had high similarity with proteins from *Hevea brasiliensis* (XP_021656809.1, 90.80%), *Jatropha curcas* (KDP29950.1, 89.61%), *Manihot esculenta* (XP_021629494.1, 90.50%), *Euphorbia helioscopia* (KP995935.1, 80.15%), and *Glycine max* (XP_003545555.1, 86.18) (Figure 1a). 

Thiolases in plant cells are divided into biosynthetic thiolases and degradative thiolases based on their functional differences [28,29]. Degradative thiolases, which are also called 3-ketoacyl-CoA thiolases and belong to thiolase I, catalyze the β-oxidation of fatty acids [30,31]. Biosynthetic thiolases, which are also called acetyl-CoA acetyltransferases and belong to thiolase II, catalyze the condensation of two acetyl-CoA to form acetoacetyl-CoA [14,15]. Among them, thiolase I is located in peroxisomes, and thiolase II is located in the cytosol [23]. Studies have shown that KAT1, KAT2 and KAT5 of Arabidopsis are located in peroxisomes, belong to thiolase I and catalyze the β-oxidation of fatty acids, ACAT1 and ACAT2 are located in the cytosol, belong to thiolase II and catalyze acetoacetyl-CoA formation [16]. In addition, the AACT proteins of *Hevea brasiliensis* and *Euphorbia helioscopia* are located in the cytosol and belong to thiolase II [19,23]. To further determine the function of EkAACT, a phylogenetic tree was constructed using the amino acid sequence of the EkAACT protein and those of AACT proteins from the species. As shown in Figure 1b, EkAACT was clustered into the same branch as acetyl-CoA acetyltransferases from *Hevea brasiliensis*, *Populus trichocarpa*, *Arabidopsis thaliana* (ACAT1 and ACAT2), *Euphorbia helioscopia*, *Glycine soja*, *Oryza sativa Japonica Group*, *Zea mays*, and *Triticum urartu*, which belong to thiolase II. *Neurospora crassa* OR74A (XP_324153.1), *Neurospora crassa* OR74A (XP_958712.1), *Arabidopsis thaliana* KAT1, *Arabidopsis thaliana* KAT2, *Arabidopsis thaliana* KAT5, and *Oryza sativa Japonica Group*, which belong to thiolase I, were grouped into the same branch. These results indicated that EkAACT belongs to cytosolic thiolase II and catalyzes two acetyl-CoA molecules to form acetoacetyl-CoA.

### 2.2. EkAACT Was Targeted to the Cytoplasm

To further determine the localization of EkAACT in cells, the pCAMBIA1300::*35S*-*EkAACT*-*GFP* recombinant expression vector was used for transient expression in tobacco (*Nicotiana benthamiana*) epidermal cells. As shown in Figure 2, compared with the positive control in which the GFP fluorescence was distributed in both the cytoplasm and the nucleus, the EkAACT-GFP fusion protein was located in the cytoplasm, which was consistent with the results of the analysis of the phylogenetic tree. 

### 2.3. Identification of Transgenic Arabidopsis

In order to study the function of *EkAACT*, Arabidopsis plants were transformed to overexpress *EkAACT*. Subsequently, six different *EkAAC**T*-overexpressing transgenic lines (T1 generation) were obtained by hygromycin (Hyg) resistance selection and named L1-L6, respectively. To confirm that the *EkAACT* gene was successfully integrated into the WT Arabidopsis genome, PCR analyses were performed by using specific primers that are able to amplify the *EkAACT* gene from the transgenic Arabidopsis genome. As shown in Figure 3, the 1098 bp bands were amplified from six different *EkAAC**T*-overexpressing transgenic Arabidopsis lines, which was consistent with the bands amplified from the pH7FWG0::*35S*-*EkAACT* recombinant expression vector as a template. However, no bands were amplified in WT Arabidopsis. The results indicated that the *EkAACT* gene was successfully integrated into WT in the Arabidopsis genome.

The relative expression levels of the *EkAACT* gene in the T2 generation of six independent transgenic and WT Arabidopsis lines were analyzed. As shown in Figure 4, the expression level of the *EkAACT* gene was different in the six different *EkAAC**T*-overexpressing transgenic lines, ranging from 4- to 40- fold compared with L1 plants. Among them, L2, L3, L4 and L6 showed significantly higher expression levels. Compared with transgenic Arabidopsis, WT Arabidopsis did not express the *EkAACT* gene. These results suggested that the *EkAACT* gene was successfully transcribed in Arabidopsis.

### 2.4. EkAACT-Overexpressing Arabidopsis Grew Better Than WT Plants

Under normal growth conditions, the L2, L3, L6 and WT seedlings were cultured on 1/2 MS medium for 10 days. The root length and whole seedling fresh weight are shown in Figure 5a,b. Compared with wild-type Arabidopsis, *EkAACT*-overexpressing transgenic Arabidopsis had longer roots and an increase in whole seedling fresh weight. The root length and whole seedling fresh weight of WT Arabidopsis were 1.56 cm and 1.92 mg, respectively. Compared with WT plants, the root length and whole seedling fresh weight of L2, L3, and L6 transgenic plants increased by approximately 21.79% and 9.20%. When the seedlings grew for 28 days, the growth morphology of Arabidopsis changed significantly. Transgenic plants grew better than WT plants (Figure 5c).

### 2.5. EkAACT Overexpression Enhanced the Expression of MVA Pathway Genes and Increased the Total Triterpenoid Accumulation

Through the observation of root length, whole seedling fresh weight and other phenotypes, we found that *EkAACT*-overexpressing Arabidopsis grew better than WT plants. RT–qPCR was used to analyze the expression level of genes related to the MVA pathway in transgenic (L2, L3, and L6) and WT Arabidopsis. The results indicated that the gene expression levels of key enzymes, including AtAACT, AtHMGS, AtHMGR, AtMK, and AtMDC, in the MVA pathway were upregulated, albeit at different levels in L2, L3, and L6 plants compared with the WT plants (Figure 6a). In addition, the total triterpenoid content of transgenic and WT plants was measured. The results indicated that the total triterpenoid content in L2, L3, and L6 transgenic plants increased by approximately 30% compared with that in WT plants (Figure 6b). 

The L2, L3, L6 and WT seedlings were cultured in a 1/2 MS medium supplemented with 50 mM NaCl and 20% PEG6000 for 10 days. Under NaCl treatments, the root length and whole seedling fresh weight of WT Arabidopsis were 0.63 cm and 1.1 mg, respectively, which were approximately 59.62% and 42.71% lower than those of WT plants under normal growth conditions. Under PEG stress treatments, the root length and whole seedling fresh weight of WT Arabidopsis were 0.59 cm and 1.02 mg, respectively, approximately 62.18% and 46.88% lower than those of WT plants under normal growth conditions. However, the transgenic Arabidopsis showed a significant increase in root length and whole seedling fresh weight under stress treatment (Figure 7a,b). Under NaCl treatment, the root length and whole seedling fresh weight of L2, L3, and L6 transgenic plants were approximately 124.34% and 34.55% higher than those of WT plants. In addition, under PEG treatment, the root length and whole seedling fresh weight of L2, L3, and L6 transgenic plants were approximately 166.67% and 69.61% higher than those of WT plants. These results indicated that the overexpression of the *Ek**AACT* gene can promote an increase in plant root length and whole seedling fresh weight under NaCl and PEG stress treatment. After treatment with 200 mM NaCl and 20% PEG solution for 30 days and 40 days, the morphology of the Arabidopsis plants was observed. Compared with those under normal conditions, salt and drought stress treatment affected the growth of both WT and *EkAACT*-overexpressing Arabidopsis, but this effect was much more severe in WT Arabidopsis. When treated with NaCl and PEG for 30 days, *EkAACT*-overexpressing Arabidopsis remained green; however, the leaves of the WT plants started to turn yellow. When treated with NaCl and PEG for 40 days, compared with transgenic Arabidopsis, the WT Arabidopsis leaves showed more serious shrinking, which indicated that *EkAACT* overexpression can enhance plant tolerance to salt and drought stress (Figure 7c).

### 2.6. EkAACT Overexpression Further Enhanced Gene Expression and Total Triterpenoid Accumulation under Abiotic Stress

In previous studies, *MsAACT*-overexpressing transgenic Medicago had a stronger tolerance to salt stress than empty vector-transformed plants, which showed that AACT, as a vital regulatory enzyme in isoprenoid biosynthesis, is involved in abiotic stress adaptation [15]. To further show that *EkAACT* overexpression can enhance plant tolerance to salt and drought by increasing the relative expression of genes in the MVA pathway and total triterpenoid accumulation, four-week-old transgenic (L2, L3, and L6) and WT Arabidopsis were treated with 200 mM NaCl and 20% PEG for two weeks to simulate salt and drought stress. Under salt and drought stress, the RT–qPCR results indicated that the expression levels of key enzyme genes, including *AtAACT*, *AtHMGS*, *AtHMGR*, *AtMK*, and *AtMDC*, in transgenic Arabidopsis were significantly higher than those in WT plants (Figure 8a,b). In addition, the total triterpenoid contents in L2, L3, and L6 transgenic plants were approximately 78.57% higher under NaCl treatment, and approximately higher by 49.13% under PEG treatment than in WT plants under stress treatment (Figure 8c,d). These results indicated that *EkAACT* gene overexpression enhances the plant tolerance to salt and drought stress by increasing relative gene expression and total triterpenoid accumulation.

### 2.7. SOD and POD Gene Expression Levels and Enzyme Activities Were Increased in EkAACT-Overexpressing Arabidopsis

Previous studies demonstrated that plants can activate a series of complex signal transmission networks to resist abiotic stress. This activation can improve tolerance to NaCl and drought stress by increasing SOD and POD enzyme activity to remove excess intracellular reactive oxygen species (ROS), which can inhibit ROS-induced intracellular component oxidation and damage [32,33]. The gene expression and enzyme activity of SOD and POD were measured to investigate whether *EkAACT*-overexpressing Arabidopsis can promote plant stress resistance to NaCl and drought. The *EkAACT*-overexpressing Arabidopsis had significantly higher *AtSOD* and *AtPOD* expression than the untreated groups (Figure 9a,b). Under normal growth conditions, the AtSOD and AtPOD enzyme activities of WT Arabidopsis were 575.541 U·g^−1^FW·min^−1^ and 358.586 U·g^−1^FW·min^−1^, respectively. Compared with WT plants, the AtSOD and AtPOD enzyme activities of L2, L3, and L6 transgenic plants increased by approximately 6.87% and 10.71%, respectively. Under NaCl stress, the AtSOD and AtPOD enzyme activities of WT Arabidopsis were 610.689 U·g^−1^FW·min^−1^ and 387.879 U·g^−1^FW·min^−1^, and those of the L2, L3, and L6 transgenic plants increased by approximately 19.91% and 15.28%, respectively. Under PEG stress, the AtSOD and AtPOD enzyme activities of WT Arabidopsis were 621.672 U·g^−1^FW·min^−1^ and 392.929 U·g^−1^FW·min^−1^, and those of the L2, L3, and L6 transgenic plants increased by approximately 19.20% and 8.48%, respectively (Figure 9c,d). These results indicated that *EkAACT*-overexpressing Arabidopsis can enhance POD and SOD enzyme activities to increase abiotic stress tolerance in plants.

### 2.8. Salt and Drought Tolerance Was Strengthened by EkAACT Overexpression

Proline is proposed to act as the main osmolyte in enhancing plant stress tolerance [34]. Proline accumulation is used as a physiological index of water shortage in plants, which increased with the decrease in the water content in plants [33]. Under NaCl and PEG treatment, the proline content of *EkAACT* transgenic Arabidopsis (L6) and wild-type plants was measured. As shown in Figure 10, WT Arabidopsis had a significantly higher proline content than transgenic L6. In addition, the proline content of wild-type and transgenic L6 was lower under normal growth conditions. According to the relationship between the proline and water content in plants, the results indicated that the water content of *Ek**AACT*-overexpressing Arabidopsis was higher than that of wild-type under salt and drought-stress treatments, and transgenic Arabidopsis showed stronger resistance to salt and drought stress.

## 3. Discussion

Plants can synthetize a variety of terpenoid metabolites that participate in the process of growth and development. Terpenoid metabolites are essential for cell functions and interactions with abiotic and biotic environments. Moreover, the ecological importance of terpenoids has gained increased attention in the development of strategies for sustainable pest control and abiotic stress protection [35]. There are two main pathways in plant terpenoid biosynthesis, namely the MVA pathway and MEP pathway [11]. In the MVA pathway, the formation of mevalonate catalyzed by HMGR has been described as the first rate-limiting step, as it is a vital synthesis step [36,37,38]. However, acetyl-CoA acetyltransferase (AACT), as a key starting enzyme, was also found to play significant roles [14]. Mutation of the Arabidopsis *AtAACT2* gene showed an embryo-lethal phenotype, which indicated that *AtAACT2* plays a significant role in isoprenoid biosynthesis [39]. Okamura et al. (2010) reported that the co-expression of *nphT7* (an alternative biosynthetic thiolase II from Streptomyces sp. strain CL190) with the *HMGS* and *HMGR* genes in *Escherichia coli* demonstrated a 3.5-fold higher production of mevalonate than when only the *HMGS* and *HMGR* genes were expressed, and this result suggests that *nphT7* can be used to significantly increase the concentration of acetoacetyl-CoA in cells [40]. Furthermore, it was found that the knockdown of *AACT* expression levels led to lower levels, altered the profiles of sterols and caused a reduced expression of downstream genes encoding FPP synthases and sterol methyltransferase [39]. Moreover, studies have shown that the overexpression of the *GlAACT* gene can also promote the transcription of related genes in the MVA pathway and the accumulation of triterpenes in *Ganoderma lucidum* [24]. These studies indicated that the activity of AACT can affect the activity of other enzymes in the MVA pathway. In this study, we showed that the overexpression of *EkAACT* in Arabidopsis remarkably upregulated the gene expression levels of *AtAACT* and *AtHMGS*, which encodes the HMG-CoA synthase at the second step, as well as *AtHMGR*, *AtMK* and *AtMDC*, which, respectively, encode the first, second and last rate-limiting enzymes in the MVA pathway (Figure 6a). Correspondingly, the gene expression level of *AACT* was also found to be strongly positively correlated with the expression levels of the HMG-CoA synthase gene [16,17]. As a result, as *AACT* and *HMGS* expression is upregulated, the concentration of HMG-CoA in cells increases, eventually promoting *HMGR* gene expression and the conversion of HMG-CoA to mevalonate. Because HMGR is the first rate-limiting enzyme and can affect the total speed of the whole metabolic pathway, *AtMK* and *AtMDC* expression levels were also enhanced, and the average total triterpenoid content of L2, L3 and L6 transgenic Arabidopsis was 30% higher than that of the wild type under normal treatment (Figure 6b). As stated above, our results proved that *EkAACT* is an important regulatory enzyme in the MVA biosynthesis pathway that can effectively increase the expression of other related genes and promote the accumulation of bioactive compounds.

The response of plants to various environmental stresses, including drought and salt stress, depends on the expression of key genes that can promote the synthesis and accumulation of bioactive compounds, which can then enhance the ability of the plant to adapt to stresses [41,42,43]. The *AACT* gene can positively respond to abiotic stresses such as salt and cold stress. Under high salt environments and cold conditions, the overexpression of the *AACT* (*MsAACT*) gene in roots and leaves can boost squalene production in Medicago, which can enhance salt and cold tolerance [15]. To further prove that the overexpression of *EkAACT* can enhance salt and drought tolerance, we treated WT and transgenic Arabidopsis with NaCl and PEG. The results showed that compared with WT plants, the root length and whole seedling fresh weight of L2, L3, and L6 transgenic plants increased by approximately 124.34% and 34.55% under NaCl treatment and increased by approximately 166.67% and 69.61% under PEG treatment (Figure 7a,b). In addition, the leaves of wild-type Arabidopsis showed faster yellowing, more severe shrinkage and faster death than those of transgenic Arabidopsis (Figure 7c). By detecting the gene expression levels of key enzymes and the total triterpenoid contents, we found that the expression levels of genes (*AtAACT*, *AtHMGS*, *AtHMGR*, *AtMK*, and *AtMDC*) in the MVA pathway and the total triterpenoid contents in *EkAACT* transgenic Arabidopsis were significantly higher than those in WT plants (Figure 8). Plants have used many mechanisms for addressing biotic and abiotic stress during their evolution. Under environmental conditions unfavorable for growth and development, plants’ internal metabolism undergoes complex changes involving the primary and secondary metabolism. For example, the secondary metabolites and terpenoids represent a wide variety of natural products in plants. Experiments have demonstrated that terpene compounds are involved in plant resistance and are believed to regulate sensitivity to biotic and abiotic stress [44,45]. In the present study, by increasing the transcription expression level of the key enzyme AACT in the terpenoid synthesis pathway, transgenic Arabidopsis increased the accumulation of total triterpenoids by approximately 78.57% under NaCl treatment, and approximately 49.13% under PEG treatment (Figure 8c,d). Consequently, the salt and drought tolerance of transgenic Arabidopsis was improved.

Under normal circumstances, physiological and other oxidative metabolic pathways in plants are in dynamic equilibrium. Under stress conditions, a series of stress reactions occur, resulting in the production of reactive oxygen species (ROS), mainly hydrogen peroxide (H_2_O_2_), hydroxyl radicals (OH), and super oxygen anions (O^2-^), in a short period of time [46]. Excess ROS poses a threat to plant cells and causes serious oxidative damage such as protein degradation. Therefore, the concentration of ROS must be kept within a certain range to be used as a signaling molecule to activate the expression of resistance genes and defense genes, promote the synthesis of plant defensive secondary metabolites, and enhance plant resistance to adverse environments as a way to avoid oxidative damage [47,48]. Under stress, the ROS response and clearance mechanisms, including enzymatic and nonenzymatic antioxidant defense mechanisms, play important roles. The enzymatic mechanisms include several well-defined systems directly involved in ROS removal such as catalase (CAT), superoxide dismutase (SOD), peroxidase (POD), and ascorbic peroxidase (APX). Nonenzymatic antioxidants include hydrophilic molecules (e.g., ascorbic), carotenoids and phenols which are the richest fat-soluble antioxidants in terpenoids in plant cells [49]. As key enzymes, SOD and POD can remove redundant ROS produced in plants under stress environments [50]. Therefore, the tolerance of plants to abiotic stress is often reflected through SOD and POD activity [51,52]. In the present study, we found that under NaCl and PEG treatment, the L2, L3 and L6 *EkAACT*-overexpressing lines upregulated the expression of the *AtPOD* and *AtSOD* genes (Figure 9a,b). Moreover, compared with WT, the AtSOD and AtPOD enzyme activities of L2, L3, and L6 transgenic plants increased by approximately 19.91% and 15.28% under NaCl stress, and increased by approximately 19.20% and 8.48% under PEG stress, respectively (Figure 9c,d). On the other hand, proline accumulation was used as a physiological index of water shortage in plants [33]. Under NaCl and PEG treatment, the increase in free proline in vivo *EkAACT*-overexpressing Arabidopsis was lower than that in WT plants (Figure 10), which indicated that *EkAACT-*overexpressing Arabidopsis has a high-water content. The results above indicated that *EkAACT-*overexpressing plants indeed had a stronger tolerance to NaCl and drought stress, which also proved that EkAACT is a regulatory enzyme of isoprenoid biosynthesis involved in abiotic stress adaptation.

## 4. Materials and Methods

### 4.1. Plant Materials

Healthy and fresh *E. kansui* leaves at the same vegetative growth periods were picked in September from the Botanical Garden of Northwest University in Shaanxi Province and used for RNA extraction in this study.

Both wild-type *Arabidopsis thaliana* (*Arabidopsis*, Brassicaceae) and *EkAACT-*overexpressing transgenic lines were derived from the Columbia (Col-0) ecotype. After surface sterilization with 75% (*v*/*v*) ethyl alcohol and 10% (*v*/*v*) HClO, the seeds were sown on 1/2 Murashige and Skoog (MS) medium supplemented with 2% (*w*/*v*) sucrose and 0.7% (*w*/*v*) agar [33] and grown in a 22 °C light incubator for 10 days. Then, the seedlings were potted in soil and grown in a greenhouse at 22 °C under a 16 h light/8 h dark illumination regime as previously described [53,54].

*Nicotiana benthamiana* growing in pots, were sown on nutrient soil of vermiculite: perlite (1:1), and grown at 24–26 °C, with a relative humidity~55% greenhouse for a 16 h day and 8 h night cycle [55].

### 4.2. Cloning and Bioinformatics Analysis of the EkAACT Gene

The coding sequence (CDS) of *EkAACT* was successfully cloned from *E. kansui* using specific primers (Appendix A) that were designed based on the *EkAACT* cDNA derived from the transcriptome database of *E. kansui* (NCBI Sequence Read Archive database SRP126436). First, according to the manufacturer’s instructions, total RNA was extracted from the fresh leaves of *E. kansui* using the Plant RNA Mini Kit (Omega, Norcross, GA, USA) and then reverse-transcribed into cDNA using oligo (dT) 18 and 5×TRUE Reaction Mix (Aidlab, Beijing, China). PCR amplifications were performed in a 25 µL final reaction volume using the above cDNA template by 5×PrimerSTAR HS DNA Polymerase (TaKaRa, Dalian, China) with an initial denaturation at 95 °C for 3 min, followed by 35 cycles at 95 °C for 30 s, 60 °C for 30 s, and 72 °C for 1 min, and a final extension at 72 °C for 10 min.

The homologous coding sequence of the *EkAACT* gene was searched and analyzed on the National Center for Biotechnology Information (NCBI) database (http://www.ncbi.nlm.nih.gov/, accessed on 15 May 2021) using online BLAST tools. The protein information (isoelectric point, molecular weight) of EkAACT was predicted by the ProtParam tool at the ExPASy website (https://web.expasy.org/protparam/, accessed on 15 May 2021). The deduced amino acid sequences of AACT proteins were used for a multiple sequence alignment analysis, which was performed with DNAMAN 6.0 software. A phylogenetic tree of the EkAACT protein and AACT proteins from other species was built with MEGA-X software version 10.2.6 by using the neighbor joining (NJ) method [56].

### 4.3. Subcellular Localization of EkAACT

The coding sequence (CDS) of *EkAACT* and *GFP* were inserted into the pCAMBIA1300 expression vector between KpnI and SacI restriction sites to generate the pCAMBIA1300::*35S*-*EkAACT*-*GFP* recombinant expression vector. The pCAMBIA1300::*35S*-*GFP* vector and pCAMBIA1300 empty vector were used in this study as a positive and negative control, respectively. Subsequently, the recombinant plasmids were transformed into the *Agrobacterium tumefaciens* strain GV3101 cells to infect tobacco epidermal cells, as described previously [57]. After 72 h of transient expression, the GFP fluorescence was recorded using a confocal microscope (Olympus, FV10-MCPUS) at 488 nm excitation.

### 4.4. Obtaining of EkAACT Transgenic Arabidopsis and Analysing of Root Length, Whole Seedling Fresh Weight and Other Phenotypes

The pH7FWG0::*35S*-*EkAACT* recombinant expression vector (Figure 11) was transformed into *Agrobacterium tumefaciens* strain GV3101 cells using the freeze–thaw method; these cells were then used to infect wild-type Arabidopsis according to previously published methods [58]. Positive transgenic seedlings were screened on 1/2 MS plates with added hygromycin (Hyg; 25 mg/L) and were confirmed by PCR in which a 1098 bp fragment of *EkAACT* was amplified from the transgenic Arabidopsis genome using specific primers (Appendix A). Then, the T3 generation seedlings were grown vertically for 10 days under 16 h light/8 h dark conditions at 22 °C and subsequently used to analyze root length and whole seedling fresh weight. Four-week-old T3 generation seedlings grown in pots were used to observe phenotypes.

### 4.5. Assay of Related Gene Expression and Total Triterpenoid Contents in Transgenic Arabidopsis

Four-week-old WT and transgenic OE*AACT*-2, OE*AACT*-3, and OE*AACT*-6 Arabidopsis lines were used to analyze the expression levels of related genes in the triterpenoid biosynthesis pathway with the primers listed in Appendix A. According to the manufacturer’s instructions, real-time qPCR was performed using SYBR Green qPCR MasterMix (Tiangen Biotech, Beijing, China), and the expression levels were calculated and analyzed with RT–qPCR using the 2DDCT method described by Livak and Schmittgen (2001) [59]. The gene *Actin 8* from Arabidopsis served as an endogenous control in the RT–qPCR experiment.

To analyze the total triterpenoid content, four-week-old WT and transgenic Arabidopsis seedlings were harvested to determine the total triterpenoid content with a colorimetric method as described previously [60]. Fresh Arabidopsis seedlings (0.4 g) were ground in liquid nitrogen using a mortar and pestle, extracted for 40 min with an ultrasonic wave (60 °C, 40 kHz) after adding 12 mL of 60% ethanol and then centrifuged at 10,000× *g* for 10 min. The supernatant was filtered and collected for testing. One milliliter of filtered solvent was evaporated in boiling water, covered and mixed after adding 0.4 mL of 5% vanillin-glacial acetic acid (5 g/100 mL) and 1.6 mL of perchloric acid; then, the mixture was hydrolyzed at 65 °C for 20 min. Eight milliliters of acetic acid was added after cooling the solution. The Oleanolic acid absorbency was measured with an ultraviolet light spectrophotometer at 550 nm, and a mixture of 0.4 mL of 5% vanillin-glacial acetic acid, 1.6 mL of perchloric acid and 8 mL of acetic acid was used as a blank control. The content of the total triterpenoids was calculated according to the Oleanolic acid standard curve.

### 4.6. Analysis of the Root Length, Whole Seedling Fresh Weight and Phenotypes of Transgenic Arabidopsis under Salt and Drought Treatments

The T3 generation seeds were sown on a 1/2 MS solid medium containing 50 mM NaCl and 20% PEG. After incubation at 4 °C in the dark for three days, the seed dormancy was broken. Then, the seedlings were grown vertically for 10 days at 22 °C in a light incubator and used to analyze root length and whole seedling fresh weight. For the phenotypic analysis, four-week-old transgenic and WT Arabidopsis plants were treated with 200 mM NaCl and 20% PEG solution once every four days for thirty and forty days, and then the growth status was recorded and imaged.

### 4.7. Assay of Gene Expression Level and Total Triterpenoid Content in Transgenic Arabidopsis under Salt and Drought Treatment

The four-week-old WT and transgenic OE*AACT*-2, OE*AACT*-3, and OE*AACT*-6 Arabidopsis lines were irrigated using 200 mM NaCl and 20% PEG once every four days for two weeks [33]. Assays of gene expression levels and total triterpenoid contents in transgenic Arabidopsis were performed as described above.

### 4.8. Expression Analysis and SOD and POD Enzyme Activity Detection under Salt and Drought Treatment

Four-week-old WT and transgenic seedlings treated as described above were used to measure the gene expression levels and activities of the SOD and POD enzymes. Transgenic and WT Arabidopsis seedlings treated with water were used as the control. According to the manufacturer’s instructions, the SOD and POD enzyme activities were measured by using the Total Superoxide Dismutase (T-SOD) Test Kit A001-1 (Jiancheng, Nanjing, China) and the peroxidase POD kit A084-3 (Jiancheng, Nanjing, China), respectively. The SOD and POD enzyme activities in the test sample were obtained using the following formula:$$\begin{array}{ll} \mathrm{SOD}\;\mathrm{activity} & \;=\;\frac{\mathrm{control}\left({\mathrm{OD}}_{550}\right)\;-\;\mathrm{determination}\left({\mathrm{OD}}_{550}\right)}{\mathrm{control}\left({\mathrm{OD}}_{550\mathrm{nm}}\right)}\\ & \;÷\;50\%\;\times \;\frac{\mathrm{total}\;\mathrm{volume}\;\mathrm{of}\;\mathrm{reaction}\;\mathrm{solution}}{\mathrm{sampling}\;\mathrm{volume}}\;÷\;\frac{\mathrm{fresh}\;\mathrm{weight}\;\mathrm{of}\;\mathrm{tissue}}{\mathrm{homogenate}\;\mathrm{volume}} \end{array}$$
$$\begin{array}{ll} \mathrm{POD}\;\mathrm{activity} & =\;\frac{\mathrm{control}\left({\mathrm{OD}}_{420}\right)-\mathrm{determination}\left({\mathrm{OD}}_{420}\right)}{12\;\times \;\mathrm{colorimetric}\;\mathrm{light}\;\mathrm{path}}\\ & \times \;\frac{\mathrm{total}\;\mathrm{volume}\;\mathrm{of}\;\mathrm{reaction}\;\mathrm{solution}}{\mathrm{sampling}\;\mathrm{volume}}\;÷\;\mathrm{reaction}\;\mathrm{time}\\ & ÷\;\frac{\mathrm{fresh}\;\mathrm{weight}\;\mathrm{of}\;\mathrm{tissue}}{\mathrm{homogenate}\;\mathrm{volume}} \end{array}$$

### 4.9. Proline Content Detection under Salt and Drought Treatment

Four-week-old WT and transgenic seedlings treated as described above were used to measure the proline content. The extraction and measurement of the free proline content were performed according to a previously described method [33,61]. The proline concentration was calculated according to a standard curve.

### 4.10. Statistical Analysis

Statistical differences between WT and transgenic plants under different treatments were analyzed using a two-tailed Student’s *t-*test with SPSS version 19.0 software. All trials were carried out in triplicate, and values are displayed as the mean ± standard error (SE). “*” and “**” indicate a significant difference at *p* ≤ 0.05 and ≤0.01, respectively.

## 5. Conclusions

EkAACT, a key initiation enzyme in the biosynthetic pathway of terpenoids, is very important for plant growth and development and plant resistance to external adverse environments. *EkAACT* overexpression can upregulate the expression of downstream genes to varying degrees to increase the accumulation of total triterpenoids. The triterpenoids accumulated in plants promote the activity of SOD and POD, which is consistent with previous studies [62], and thereby enhances plant tolerance to abiotic stress (Figure 12).

## Figures and Tables

**Figure 1 plants-11-01539-f001:**
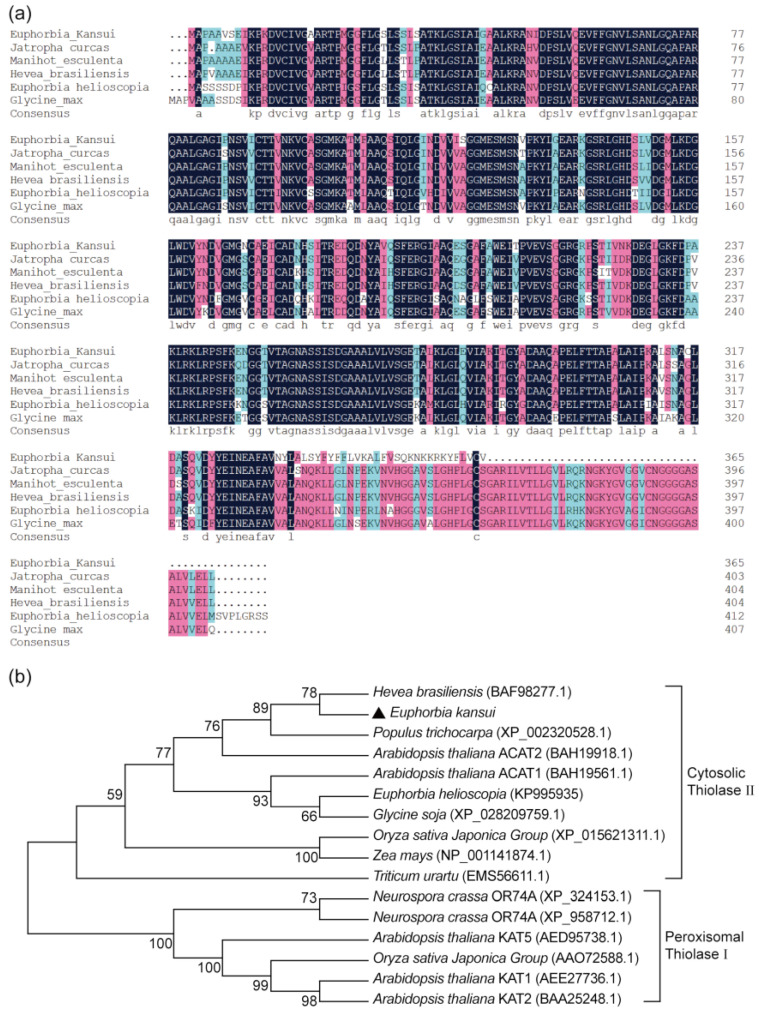
Bioinformatics analysis of the *EkAACT* gene. (**a**), Homology analysis of the EkAACT protein. The deduced amino acid sequences of EkAACT and AACT proteins from other species were aligned with DNAMAN software. Dark blue background: similarity is 100%, pink background: similarity is ≥75%, and light blue background: similarity is ≥50%. (**b**), Phylogenetic tree of EkAACT homologous proteins created by MEGA-X using the maximum likelihood method. The bootstrap value was 1000, and the branch lengths were calculated by the Poisson model.

**Figure 2 plants-11-01539-f002:**
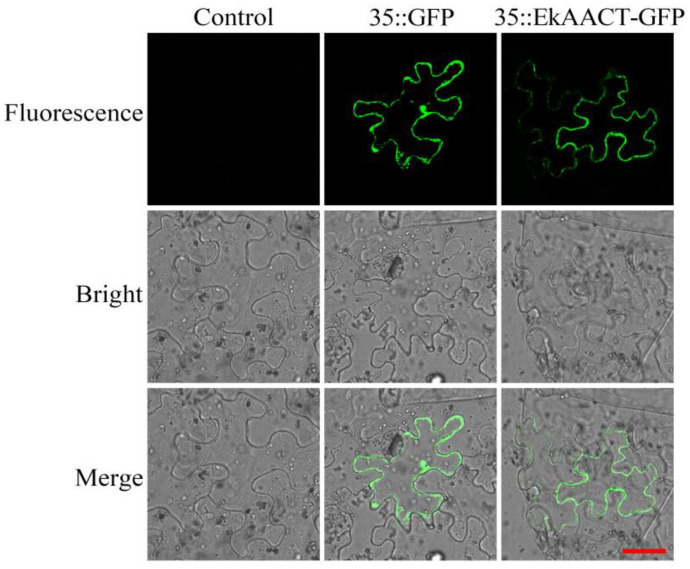
Subcellular localization of EkAACT protein in tobacco epidermal cells. Control: 1300 empty vector transformed in tobacco epidermal cells. Bar = 40 µm.

**Figure 3 plants-11-01539-f003:**
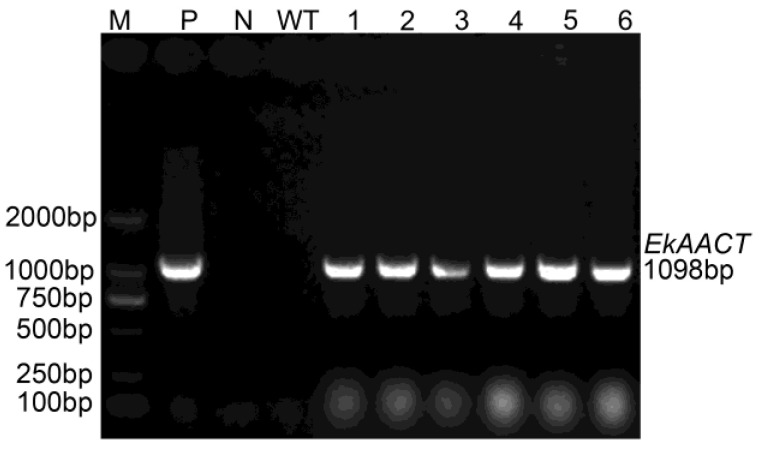
The *EkAAC*T gene was detected in six transgenic lines. Transgenic seedling detection of *EkAACT*-overexpressing Arabidopsis using PCR. Abbreviations: M: D_2000_ marker; N: water (negative control); P: pH7FWG0::*35S*-*EkAACT* vector (positive control); WT: wild-type Arabidopsis; and 1–6: L1–L6 transgenic Arabidopsis.

**Figure 4 plants-11-01539-f004:**
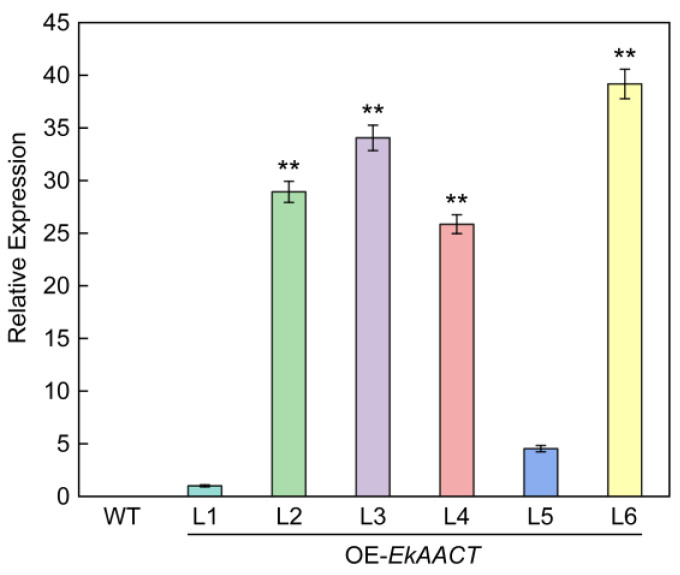
Detection of *EkAACT* gene expression levels in Arabidopsis by using RT-qPCR. Expression levels of genes were represented as the expression relative to a Line 1 that was set to 1.0. WT: wild-type Arabidopsis; L1–L6: *EkAACT*-overexpressing Arabidopsis lines. All data are the mean ± SE of three biological replicates. “**” indicates a significant difference from that of the WT at *p* ≤ 0.01 by Student’s *t*-test.

**Figure 5 plants-11-01539-f005:**
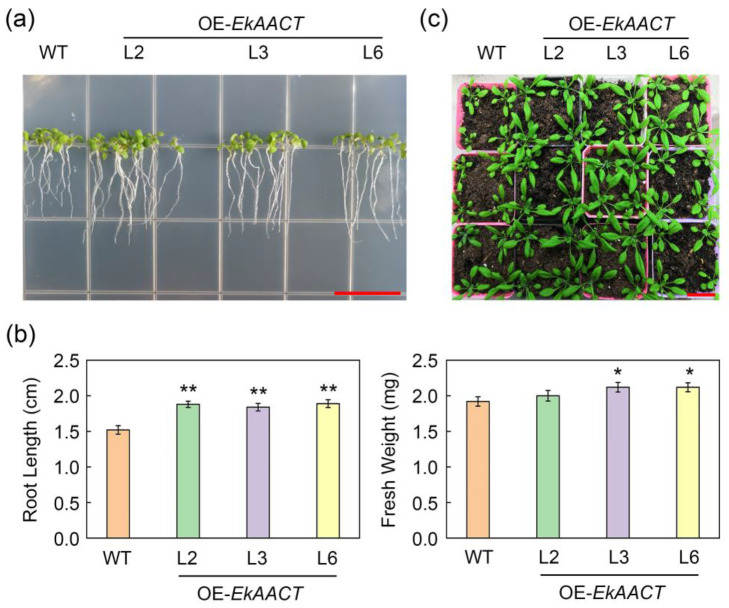
Overexpression of *EkAACT* promotes plant growth. (**a**,**b**), Analysis of root length and whole seedling fresh weight in 10 days old Arabidopsis under a normal environment. Bar = 15 mm. (**c**), Photographs of four-week-old Arabidopsis under a normal environment. Bar = 30 mm. WT: wild-type Arabidopsis; L2, L3 and L6: transgenic Arabidopsis lines. Root length and fresh weight data are presented as the mean ± SE of ten biological replicates. “*” and “**” indicate a significant difference from that of the WT at *p* ≤ 0.05 and ≤0.01, respectively, by Student’s *t*-test.

**Figure 6 plants-11-01539-f006:**
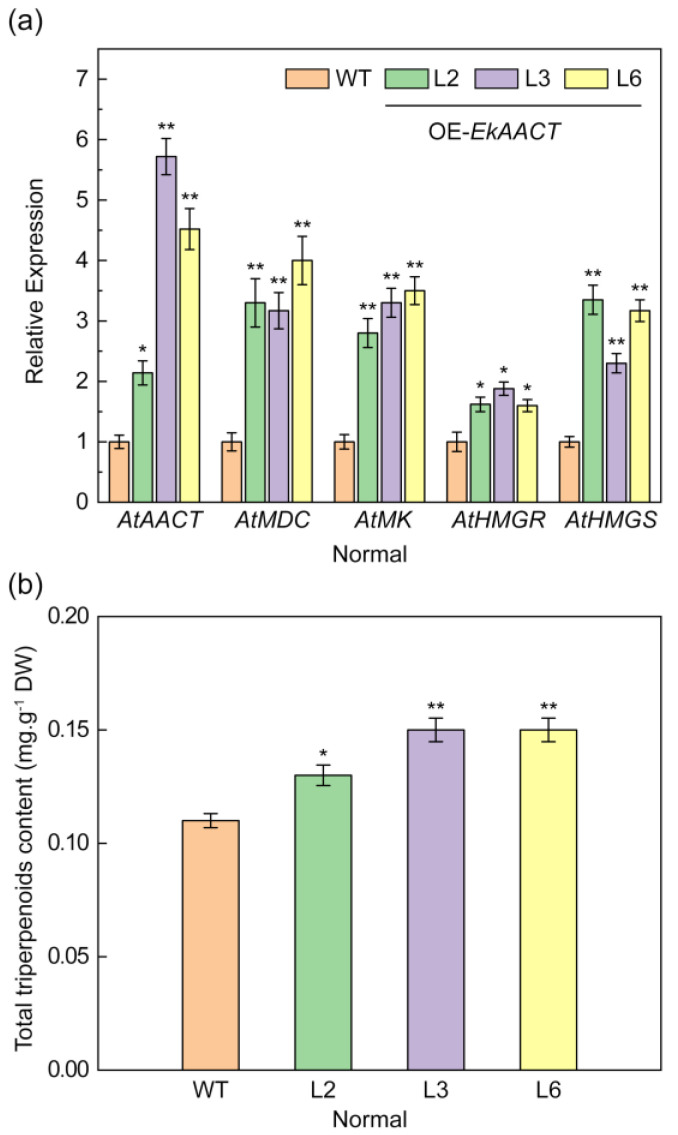
*EkAACT* overexpression promotes the expression of genes related to the mevalonate pathway (MVA) and triterpene accumulation. (**a**), Expression level detection of related genes in the MVA pathway under a normal growth environment. Expression levels of genes are represented as expression relative to a WT that was set to 1.0. (**b**), Detection of the total triterpenoid content in a normal growth environment. WT: wild-type Arabidopsis; L2, L3 and L6: transgenic Arabidopsis lines. All data are the mean ± SE of three biological replicates. “*” and “**” indicate a significant difference from that of the WT at *p* ≤ 0.05 and ≤0.01, respectively, by Student’s *t*-test.

**Figure 7 plants-11-01539-f007:**
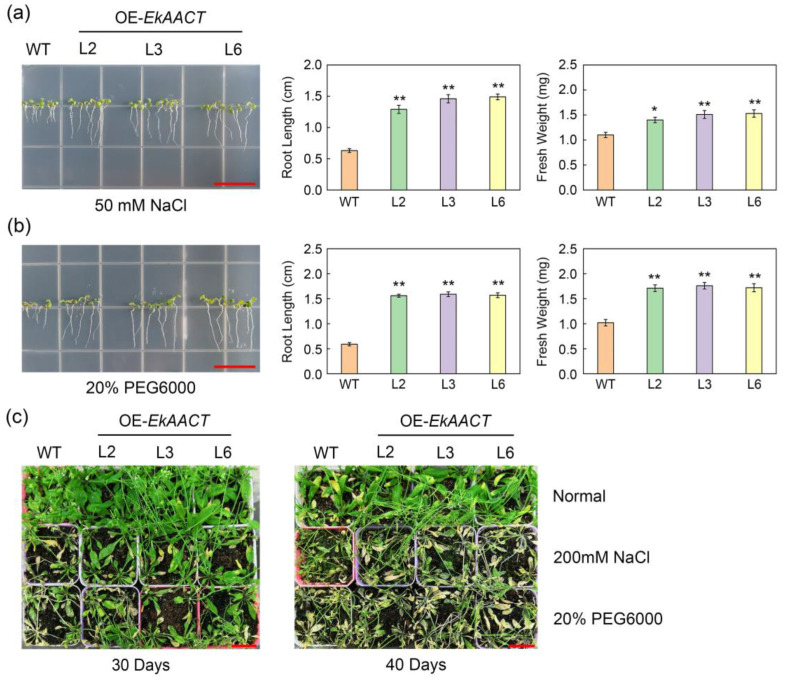
Overexpression of *EkAACT* causes plants to grow better under stress. (**a**,**b**) Changes of root length and whole seedling fresh weight in 10 days old Arabidopsis grown on 1/2 MS solid medium containing 50 mM NaCl and 20% PEG. Bar = 15 mm. (**c**) Photographs of four-week-old Arabidopsis under 200 mM NaCl and 20% PEG treatment for 30 and 40 days. Bar = 30 mm. WT: wild-type Arabidopsis; L2, L3 and L6: transgenic Arabidopsis lines. Root length and fresh weight data are presented as the mean ± SE of ten biological replicates. “*” and “**” indicate a significant difference from that of the WT at *p* ≤ 0.05 and ≤0.01, respectively, by Student’s *t*-test.

**Figure 8 plants-11-01539-f008:**
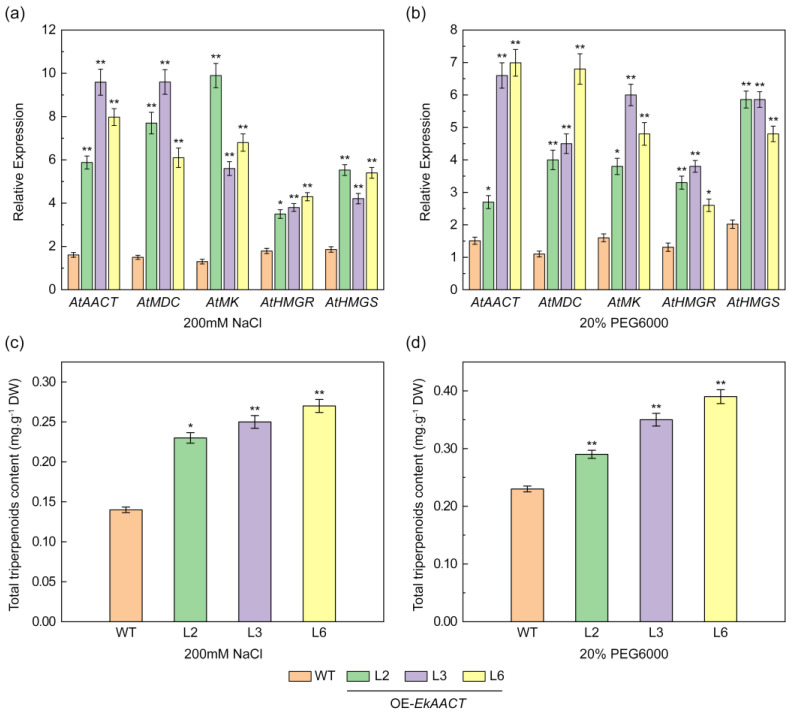
*EkAACT* overexpression further promotes the expression of genes related to the mevalonate pathway (MVA) and triterpene accumulation under stress. (**a**,**b**), Expression level detection of related genes in response to 200 mM NaCl and 20% PEG treatment. Expression levels of genes are represented as expression relative to a WT that was set to 1.0. (**c**,**d**), Changes in the total triterpenoid contents in response to stress treatment. WT: wild-type Arabidopsis; L2, L3 and L6: transgenic Arabidopsis lines. All data are the mean ± SE of three biological replicates. “*” and “**” indicate a significant difference from that of the WT at *p* ≤ 0.05 and ≤0.01, respectively, by Student’s *t*-test.

**Figure 9 plants-11-01539-f009:**
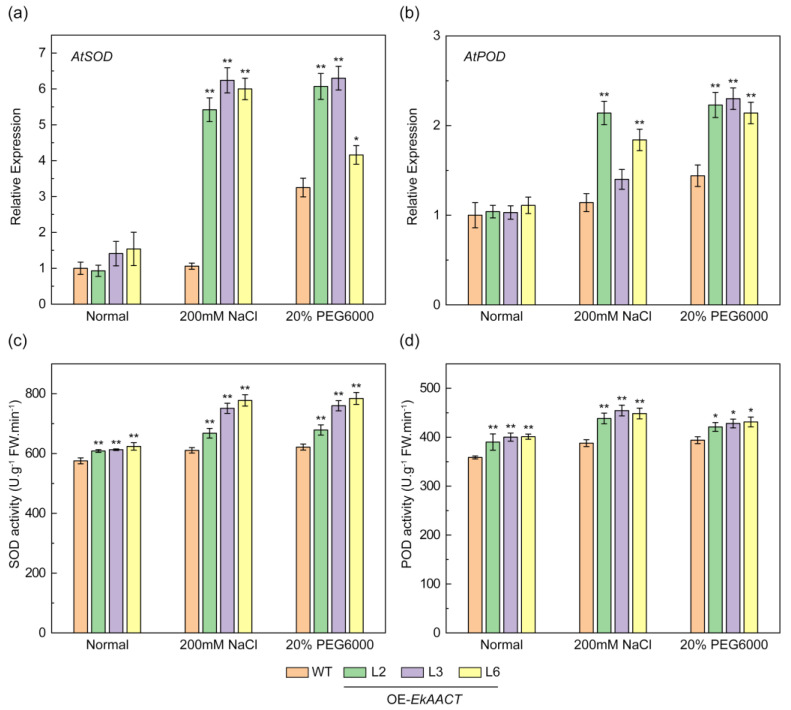
*EkAACT* overexpression enhances *At**SOD* and *At**POD* gene expression and enzyme activities under stress. (**a**,**b**), *At**SOD* and *At**POD* expression analysis. Expression levels of genes are represented as the expression relative to a WT that was set to 1.0. (**c**,**d**), Identification of AtSOD and AtPOD enzyme activities. WT: wild-type Arabidopsis; L2, L3 and L6: transgenic Arabidopsis lines. All data are presented as the mean ± SE of three biological replicates. “*” and “**” indicate a significant difference from that of the WT at *p* ≤ 0.05 and ≤0.01, respectively, by Student’s *t*-test.

**Figure 10 plants-11-01539-f010:**
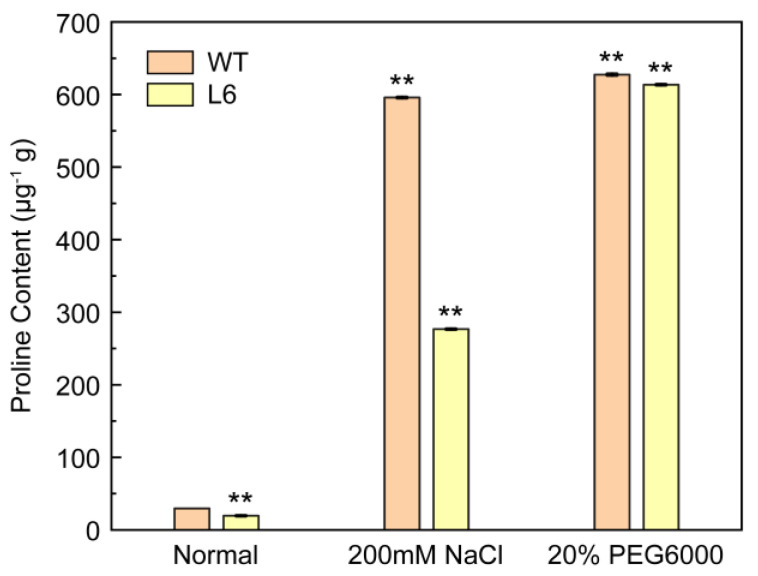
*EkAACT* overexpression decreases the proline content in plants under abiotic stress. WT: wild-type Arabidopsis; L6: transgenic Arabidopsis Line 6. All data are presented as the mean ± SE of three biological replicates. “**” indicate a significant difference from that of the WT at *p* ≤ 0.05 and ≤0.01, respectively, by Student’s *t*-test.

**Figure 11 plants-11-01539-f011:**
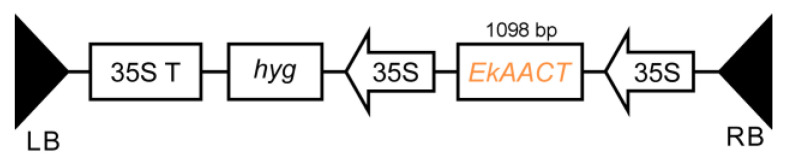
Schematic diagram of the T-DNA region of the binary plasmid pH7FWG0::*35S*-*EkAACT*. *LB*, left border; *RB*, right border; *35S T*, cauliflower mosaic virus (CaMV) 35S terminator; *hyg*, Streptomyces hygroscopic gene; and *EkAACT*, *Euphorbia kansui* acetyl-CoA acetyltransferase gene.

**Figure 12 plants-11-01539-f012:**
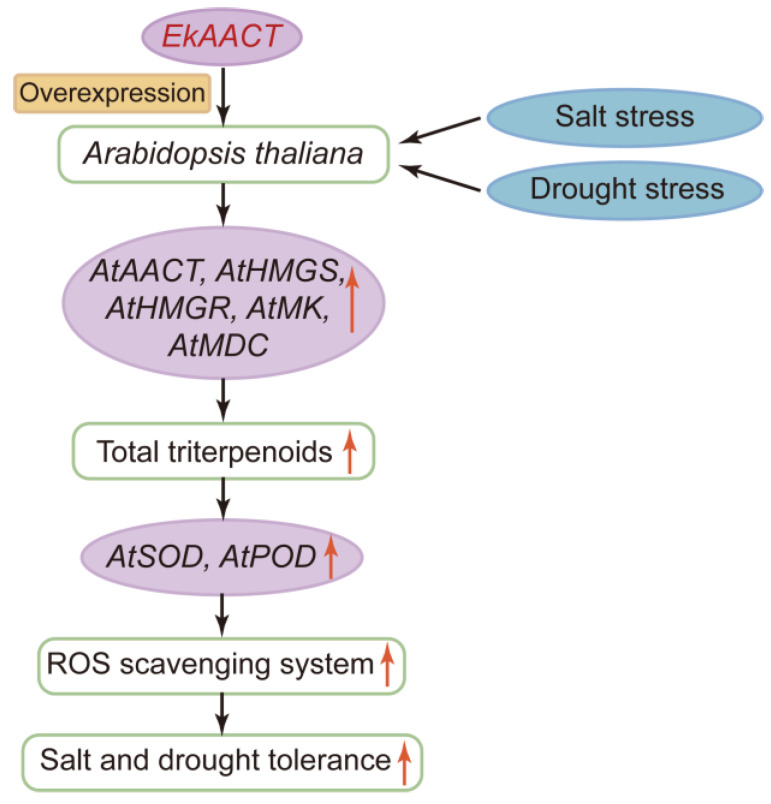
A proposed model for the regulatory network of *EkAACT* overexpression in response to abiotic stress. Genes involved in triterpenoid biosynthesis were upregulated in *Ek**AACT*-overexpressing Arabidopsis under treatment with NaCl and PEG, which promoted triterpenoid accumulation. Moreover, the increase in total triterpenoids enhanced the expression of the *At**SOD* and *At**POD* genes in the ROS scavenging pathway and strengthened the activity of the SOD and POD enzymes. As a result, the salt and drought tolerance of *Ek**AACT*-overexpressing Arabidopsis was enhanced because of the inhibition of ROS formation.

## Data Availability

The data that support the findings of this study are available from the corresponding authors upon reasonable request.

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
