# Peer review of "Molecular Cloning and Analysis of an Acetyl-CoA C-acetyltransferase Gene (EkAACT) from Euphorbia kansui Liou"

_plants, 2022, doi:10.3390/plants11121539_

Round 1
Reviewer 1 Report
In this manuscript, the authors characterized an acetyl-CoA C-acetyltransferase gene (EkAACT) from Euphorbia kansui. After analyzing the sequence and phylogeny, the protein was overexpressed heterogeneously in Arabidopsis thaliana plants. The phenotype of the plants overexpressing EkAACT was analyzed to characterize the function of EkAACT in plant development and abiotic stress tolerance. Although the data are interesting and are presented in a logical order, I think that several points need to be addressed. Please see below my detailed comments and suggestions.
The writing of the manuscript should be improved. Indeed, numerous English errors are present in the manuscript, so I suggest the authors to use an editing service.
Line 113-114: The results only « suggest » that EkAACT catalyzes the formation of acetoacetyl-CoA. To properly demonstrate that EkAACT is capable of this specific enzymatic activity, enzymatic assays should be realized. Moreover, the subcellular localization of the protein is speculated from the phylogenetic analysis, but experiments should be conducted to prove that EkAACT is indeed cytosolic.
Line 49: “fresh weight”. It should be explained if the fresh weight corresponds to the whole seedling.
Line 116: First of all, it should be explained that, in order to study the function of EkAACT, Arabidopsis plants were transformed to overexpress EkAACT.
Line 117: “Hyg” if an abbreviation is needed, it should be explained.
Line 129: “positive control” should be explained in the legend:
pH7FWG0::35S-EkAACT vector (positive control)
Line 127: “positive seedling detection” is not appropriate and should be reformulated.
Line 206: The age of the plants and the duration of treatment should be mentioned in the figure legend for the Figure 7a, b, and c.
Line 149: “higher moisture content”. To support this statement, data should be provided.
Figures: A scale bar should be indicated for each photograph.
Line 150: “The root length and fresh weight of WT Arabidopsis were 1.56 cm and 1.92 mg, respectively.” The photograph should be consistent with the measurements. It looks to me that the photograph (Figure 5 a) was perhaps taken on younger plants than the plants used for the measurements. Is it the case or not?
Since the tolerance to abiotic stress is dependent on the terpenoid hormone ABA, did the authors checked if the ABA level is changed in the transgenic lines compared to the wild-type plants? Did the authors check the total tetraterpenoid and antioxidant pigment content as well?
Line 275: “the water content of EkAACT-overexpressing Arabidopsis was higher than that of wild-type under salt and drought stress treatments”. The conclusion should be reformulated to explain that the water content was estimated through the proline content.
Line 473: the formula should be explained.
Line 488: How do the triterpenoids promote the activity of SOD and POD? A reference should be provided to support this conclusion.
Author Response
Dear reviewer:
I have revised the questions raised by the reviewers item by item. Please see the attachment for the results

Reviewer 2 Report
This is an interesting study, however there are a few issues that should be corrected prior to publication:
- The first figure I see is figure 2. The authors should renumber the figures starting with 1. Right now Figure 1 is in between figures 10 and 11, which is confusing.
- Figures 5 and 7 should include scale bars with the pictures so the size of the plants can be easily determined.
Overall a nice study.
Author Response
Dear reviewer:
I have revised the questions raised by the reviewers item by item. Please see the attachment for the results.

Round 2
Reviewer 1 Report
Authors answered all the queries and improved tge manuscript